# Review of Degradation Mechanism and Health Estimation Method of VRLA Battery Used for Standby Power Supply in Power System

**Ruxin Yu [1], Gang Liu [1], Linbo Xu [1], Yanqiang Ma [2], Haobin Wang [2,*] and Chen Hu [3,\*]**

[1] Zhejiang Zheneng Jiahua Electric Power Generation Co., Ltd., Jiaxing 314201, China
[2] Hebei Chuangke Electronic Technology Co., Ltd., Handan 056107, China
[3] Laboratory of Operation and Control of Renewable Energy & Storage Systems,
    China Electric Power Research Institute, Beijing 100192, China
* Correspondence: whbwag@163.com (H.W.); whhuchen@163.com (C.H.)

**Abstract:** As the backup power supply of power plants and substations, valve-regulated lead-acid (VRLA) batteries are the last safety guarantee for the safe and reliable operation of power systems, and the batteries' status of health (SOH) directly affects the stability and safety of power system equipment. In recent years, serious safety accidents have often occurred due to aging and failure of VRLA batteries, so it is urgent to accurately evaluate the health status of batteries. Accurate estimation of battery SOH is conducive to real-time monitoring of single-battery health information, providing a reliable guarantee for fault diagnosis and improving the overall life and economic performance of the battery pack. In this paper, first, the floating charging operation characteristics and aging failure mechanism of a VRLA battery are summarized. Then, the definition and estimation methods of battery SOH are reviewed, including an experimental method, model method, data-driven method and fusion method. The advantages and disadvantages of various methods and their application conditions are analyzed. Finally, for a future big data power system backup power application scenario, the existing problems and development prospects of battery health state estimation are summarized and prospected.

**Keywords:** backup power supply; VRLA batteries; aging failure mechanism; state of health; evaluation methods

## 1. Introduction

Direct current (DC) power supply systems play a very important role in power plants and substations. VRLA batteries are widely used as backup power to ensure normal operation of power plants and substations. Once alternating current (AC) suddenly loses power, the VRLA batteries immediately start to supply power to important DC load equipment, such as relay control and protection devices, automation devices, opening and closing mechanisms of high-voltage circuit breakers, communication equipment, emergency lighting lamps, etc. [1,2]. The abnormal failure of a VRLA battery as the backup power supply for a DC power supply system, seriously affects the safe operations of the DC power supply system. Therefore, the battery is considered the most core component of the DC power supply system, and it is an important guarantee for the safe and stable operation of power plants and substation systems [3].

In recent years, major accidents have occurred in power plants and substations due to VRLA battery failures, which has caused considerable security risks to the safe and stable operation of power grids. In 2013, AC an circuit was out of power due to a lightning strike at a substation of a power grid company in China. Due to the failure of the battery pack, some circuit breaker switches could not work normally, which eventually led to a serious accident of voltage loss in the whole substation. Cause analysis revealed that some

negative plate straps of VRLA batteries in the substation were seriously corroded. When faced with heavy load impact, the output voltage of the battery pack dropped considerably and therefore could not meet the requirement of the minimum voltage of switching action. As a result, some switches could not work normally, and the breakdown could not be isolated in time, which eventually led to total substation voltage loss [4]. In June 2016, a fire broke out due to equipment failure, and the lead-acid battery failed to supply power in time due to failure, resulting in an accident of voltage loss in a 330 kV substation in the western suburb of Xi'an, China [4]. Similar serious accidents due to VRLA failure have also occurred in substations in other areas [5,6].

Therefore, analysis of the causes of battery failure and estimation and prediction of the state of health (SOH) of batteries are helpful to identify malfunctioning batteries in a timely manner and make maintenance plans. This is of great significance for prolonging the service life of the battery, reducing the maintenance cost of the system and ensuring the safe operation of the power system.

Many studies have been conducted on the SOH of lithium-ion power battery [7], but few comprehensive reviews have been conducted on the SOH of batteries for standby power supply [8]. Cuma et al. [9] conducted a comprehensive review of various estimation strategies used in hybrid and battery electric vehicles, focusing on battery fault diagnosis, state of charge (SOC) and state of health (SOH) estimation. However, in this study included few applications involving lead-acid batteries. Waltari et al. [10] introduced fault classification and state-of-health monitoring methods for lead-acid batteries. Battery failures are classified into three categories: high impedance, low impedance (short circuit) and deterioration of capacity. Battery health monitoring methods including string-voltage-based, cell-voltage-based, current-based and impedance-based methods were reviewed. However, the article was published in 1999, and the reviewed methods of battery health monitoring are simple and outdated. Ouyang et al. [11] reviewed capacity forecasting technology for VRLA batteries. They divided the capacity forecasting methods into open-circuit voltage measuring, Coulomb counting and internal resistance methods. However, the study did not highlight the influencing factors of SOH.

In view of the lack of summary of the estimation approaches to estimate SOH for VRLA batteries for standby power supply, a detailed and up-to-date summary is necessary. In this paper, the research progress of the decay mechanism of VRLA batteries and the method of estimating SOH are reviewed. First, we introduce the working mode and failure mechanism of the standby VRLA floating charge mode. Then, we describe the principles of SOH estimation methods, practical application cases and the advantages and disadvantages of these estimation methods. This review can provide a decision-making basis for the operation, maintenance and scientific management of standby power supply.

The remainder of this paper is arranged as follows: Section 2 summarizes the floating charge operation characteristics and failure mechanism of backup VRLA batteries. Section 3 introduces the definition of battery SOH. Section 4 introduces the classification and characteristics of different SOH estimation methods for VRLA batteries. Finally, in Section 5, the conclusions are summarized.

## 2. Operation Characteristics and Failure Mechanism of VRLA Floating Charge

### 2.1. Operating Characteristics of Standby VRLA

The operation mode of a valve-regulated lead-acid battery for standby power supply includes initial charging before operation, floating charging in normal operation, balanced charging every three months, emergency power supply in case of AC interruption, constant current and constant voltage charging after AC recovery, etc., as shown in Figure 1. VRLA is in the cycle mode of floating charge and equalizing charge for a long time. Once AC power is lost, VRLA is used as the standby DC emergency power supply. After the AC power returns to normal, the charger charges the battery with constant current and voltage. Then, VRLA switches to floating charge and equalizing charge cycle mode [12].

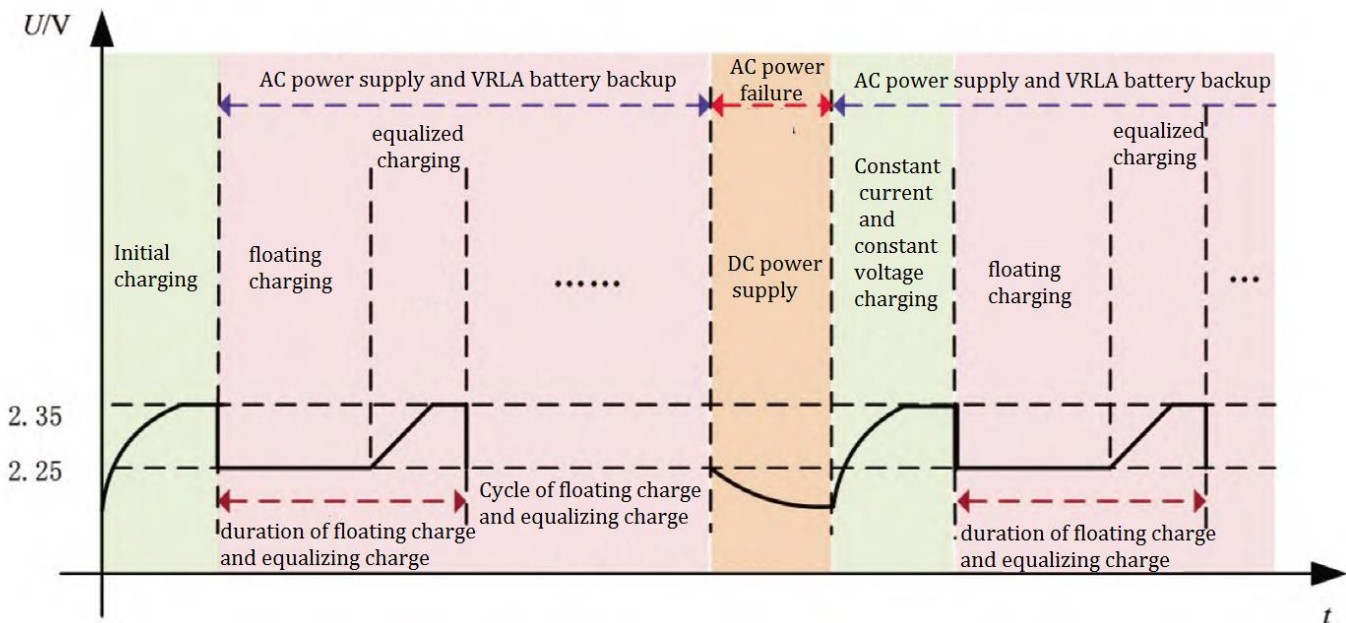

**Figure 1.** Charging and discharging operation mode diagram of a VRLA battery [12]. Adapted with permission from Ref. [12]. 2022, Chinese Journal of Power Sources magazine.

### 2.2. Aging Mechanism of VRLA in Floating Charge Operation

VRLA is widely used in the power industry because it has the advantages of low price, mature technology, safety, reliability and easy maintenance. In power plants and substations, VRLA operates in floating charge mode for a long time. Various processes promote the aging of VRLA batteries, such as positive grid corrosion, irreversible sulfation, softening of positive active material, negative plate strap corrosion, water loss, etc., as shown in Table 1 [13,14].

**Table 1.** VRLA battery failure modes.

| Failure Mode | Failure Reason | Phenomena |
| --- | --- | --- |
| Positive grid corrosion | In the environment of strong acid, strong oxidation and high potential, the positive grid alloy is thermodynamically ustable, and oxidation corrosion is inevitable | Capacity reduction and ncrease in internal resistance |
| Irreversible sulfation of negative electrode | When the floating charge voltage is too low, $PbSO_4$ crystals with coarse particles and poor chemical activity are formed on the negative electrode surface. | The battery capacity is significantly reduced; the voltage rises quickly when charging and drops rapidly when discharging |
| Softening and shedding of positive active material | In the positive active substance, the composite structure of $\alpha$-$PbO_2$/$\beta$-$PbO_2$ crystal and $PbO_2$-$PbO(OH)_2$ gel is destroyed, which leads to a decrease in the binding force between active substance particles | During the initial stage of use, the battery capacity is reduced |
| Corrosion of negative plate strap | The metal lead of the negative bus bar is slowly corroded over a long time and transformed into powder $PbSO_4$ crystal | The open-circuit voltage and floating charge voltage are low, and the internal resistance is high |

**Table 1.** *Cont.*

| Failure Mode | Failure Reason | Phenomena |
|---|---|---|
| Battery leakage | The battery is not tightly sealed or the shell is broken | Sulfuric acid leakage, pole corrosion and pole temperature rise |
| Thermal runaway | High voltage and current cause a large amount of heat to accumulate in the battery, causing the battery temperature to rise rapidly | Increase in the floating charge current, temperature rise and battery swelling |

(1)  Corrosion of positive grid

Corrosion of the positive grid is one of the most common failure modes of VRLA batteries, which refers to the process by which the lead alloy of the positive grid is oxidized to lead dioxide. Under strong oxidation and a high-potential environment, the thermodynamic instability of lead and lead alloys is the fundamental cause of positive grid corrosion. At the end of charging, the positive grid is usually in the potential range of 1.3~1.4 V, which is much higher than the protection potential of lead alloy. The following electrochemical reactions occur:

$$Pb + 2H_2O \rightarrow PbO_2 + 4H^+ + 4e^- \tag{1}$$

In the case of overcharge, the acidity near the positive plate increases due to oxygen evolution reaction [15,16]. The composition of the grid alloy is the main factor affecting the corrosion rate of the positive plate, in addition to environmental temperature, floating charge voltage, casting process and other factors [17]. Corrosion of the positive plate may reduce the mechanical strength of the grid, break the grid, increase the ohmic internal resistance and rapidly increase the voltage during charging [18].

(2)  Irreversible sulfation of the negative electrode

The main active substance of a VRLA battery cathode is sponge lead. During discharge, spongy Pb is converted into crystal $PbSO_4$, and $PbSO_4$ is reversibly converted into Pb during charging [19]. When the battery is in the state of deep discharge, undercharge, open-circuit or low-rate discharge for a long time, the $PbSO_4$ crystal of the battery anode cannot be completely converted into spongy Pb [20]. The coarse $PbSO_4$ crystal gradually covers the negative plate surface, and the inert $PbSO_4$ no longer participates in the chemical reaction, that is, irreversible sulfation. Irreversible sulfation affects the recombination of $H_2$ and $O_2$ into $H_2O$ in the battery, resulting in the inability of active substances in the electrode plate to participate in the reaction, which increases the battery internal resistance and causes premature battery failure [21].

(3)  Negative plate strap corrosion

D. Pavlov [22] et al. thought that the oxygen generated by the positive electrode partially gathered at the upper part of the electrode group, which caused the negative electrode tab and bus bar to lose cathodic protection. If the anode tab and bus bar are farther away from $AGM/H_2SO_4$ system, the potential of the anode bus bar is higher than the equilibrium potential of $PbSO_4/Pb$, and metallic lead slowly corrodes and transforms into powdered $PbSO_4$ crystal [23]. When the corrosion is serious, the surface and even the inside of the bus bar are seriously pulverized, resulting in the reduction in its mechanical strength. Under the action of stress, the bus bar breaks, resulting in the failure of the battery due to the open circuit inside.

(4)  Softening and shedding of positive active material

During charging and discharging, the structure of the positive active material of the battery is damaged, which leads to a reduction in the binding force between the active material and the grid and ultimately leading to the active material falling off. Pavlov

believed that in a gel–crystal system, with the charge–discharge cycle, the oxygen evolution reaction in the battery destroys the polymer chain in PAM (positive active material), resulting in an increase in the crystallinity of PAM. D Pavlov thought that the positive active material was a gel–crystal system and that its smallest active material unit was composed of $\alpha$-$PbO_2$/$\beta$-$PbO_2$ crystal and $PbO_2$-$PbO(OH)_2(OH)_2$ gel, which were in a state of mutual balance. With the charge–discharge cycle, the amorphous state in the active material gradually crystallizes, and the crystal area with poor binding force increases. This reduces the binding force between the active material units and ultimately leads to the softening and shedding of the positive active material [24,25].

Zhong et al. [26] classified the main failures of VRLA batteries during floating charging into three categories: positive grid corrosion, negative busbar corrosion and negative sulfation. The positive grid first undergoes electrochemical corrosion, which intensifies the oxygen evolution reaction on the anode. The large amount of oxygen released from the positive electrode increases the oxygen recombination reaction at the negative electrode of the battery and intensifies the corrosion of the negative busbar. At the same time, the process of anode grid corrosion and oxygen release requires water consumption, which increases the oxygen transmission channel. The oxygen recombination reaction is further intensified inside the battery, which also causes the risk of thermal runaway of the battery. In addition, there are other failure modes, such as micro-short circuit, shell rupture, etc. [27]. The aging mechanism of a VRLA battery is often dominated by one failure mode, and the others coexist and interact with each other. Therefore, the degradation of VRLA battery capacity is the result of the interaction of various aging factors [14].

### 2.3. Aging Mechanism Analysis Method

The methods for analyzing the aging mechanism of batteries can be divided into three categories, namely external characteristic analysis (electrical testing), disassembly analysis and in situ online analysis [28]. External characteristic analysis, such as charge–discharge curve and electrochemical impedance spectroscopy (EIS), incremental capacity analysis (ICA), differential voltage analysis (DVA), etc., are used to extract the aging characteristics of the battery by properly processing the external characteristics (voltage, capacity, internal resistance, etc.) of the battery. The disassembly analysis method is also called material physical and chemical property testing and analysis. First, the aged battery is disassembled in a suitable environment to determine the internal materials of the battery, including grids, separators, positive and negative active substances, electrolytes, etc. Then, these materials are tested by analytical techniques such as scanning electron microscope (SEM), X-ray diffraction technology (XRD) and inductively coupled plasma technology (ICP) to obtain material information such as microscopic morphology, crystal structure and element distribution [29]. The in situ online analysis method involves the use of in situ analysis equipment to monitor the change of the internal physical characteristics of the battery during the cycle and analyze the evolution of the internal material of the battery during the aging process.

The advantages and disadvantages of the three methods for analyzing the aging mechanism of batteries are compared in Table 2, and the main test technologies in each analysis method are listed. The disassembly analysis method and in situ test technology method usually require expensive experimental equipment and cannot analyze the aging behavior of the battery online, so their application is limited. The external characteristic analysis method is based on the battery charge and discharge or impedance spectrum to analyze and extract the aging characteristics of the battery without damage to the battery sample, so it is suitable for online estimation of the aging behavior of the battery.

**Table 2.** Comparison of three methods for analyzing aging mechanisms of batteries.

| Aging Analysis Method | Pros | Cons | Testing Technique |
|---|---|---|---|
| External characteristic analysis | The studied battery is not damaged; the evolution of battery aging at different life stages can be studied | The aging mechanism is analyzed based on speculation and needs to be verified by the disassembly analysis method | ICA, DVA, EIS |
| Disassembly analysis | The physical and chemical properties of the internal materials of the battery can be directly characterized; internal causes of aging can be determined, and different failure modes can be distinguished | The studied battery is inevitably damaged | SEM, XRD, ICP |
| In situ online analysis | The studied battery is not damaged; the evolution of the material inside the battery is characterized in situ at different life stages | Requires complicated devices | In situ XRD, neutron diffraction |

A VRLA battery is a complex electrochemical system, and its capacity decay is nonlinear. The aging mechanism of a battery is complex and is influenced by many factors. The analysis of battery decay failure mechanisms is helpful to determine the health factors that can best characterize battery SOH. Battery performance is tested by a variety of testing technologies to detect the battery failure mode and obtain aging information. On this basis, combined with various models, the battery SOH can be accurately predicted to ensure the safe operation of the battery.

## 3. Definition of Battery SOH

SOH represents the ability of a current battery to store electric energy compared with that of a new battery. With the increase in service time, the internal resistance of the battery increases, and the maximum usable capacity decreases. Therefore, capacity and internal resistance parameters are often used to define battery SOH in the industry.

According to the definition in terms of capacity, the SOH can be expressed as

$$\mathrm{SOH} = \frac{C_{curr}}{C_{rated}} \times 100\% \tag{2}$$

where $C_{rated}$ is the nominal capacity, and $C_{curr}$ is the present maximum available capacity, which can be measured by discharging the battery at a fixed current (usually 0.1 $C_{rated}$) and air temperature (usually 20 °C to 30 °C).

According to the definition in terms of resistance, the SOH can be expressed as

$$\mathrm{SOH} = \frac{R_{EOL} - R_{curr}}{R_{EOL} - R_{new}} \times 100\% \tag{3}$$

where $R_{new}$ represents the initial internal resistance of the new battery, $R_{curr}$ represents the actual internal resistance under the current cycle and $R_{EOL}$ represents the internal resistance at the end of the battery life [30]. Another parameter used to describe the state of a battery is the SOC, which is defined as

$$\mathrm{SOC} = \frac{C_{remain}}{C_{curr}} \times 100\% \tag{4}$$

where $C_{remain}$ is the remaining capacity of the battery. According to Formulas (2) and (4), SOC and SOH are closely linked through $C_{curr}$. Therefore, the accurate estimation of SOH must be related to SOC.

According to the IEEE 1188-2005 standard, when the actual capacity of a VRLA is less than 80% of the rated capacity, that is, the SOH is less than 80%, the battery must be maintained or replaced [31].

## 4. SOH Estimation Methods

At present, battery SOH estimation methods include the experimental method, the model method, the data-driven method and the fusion method. In this study, the different SOH estimation methods are classified into four different categories: experimentally based, model-based, data-driven and fusion methods. Figure 2 shows the classification of the different SOH estimation methods.

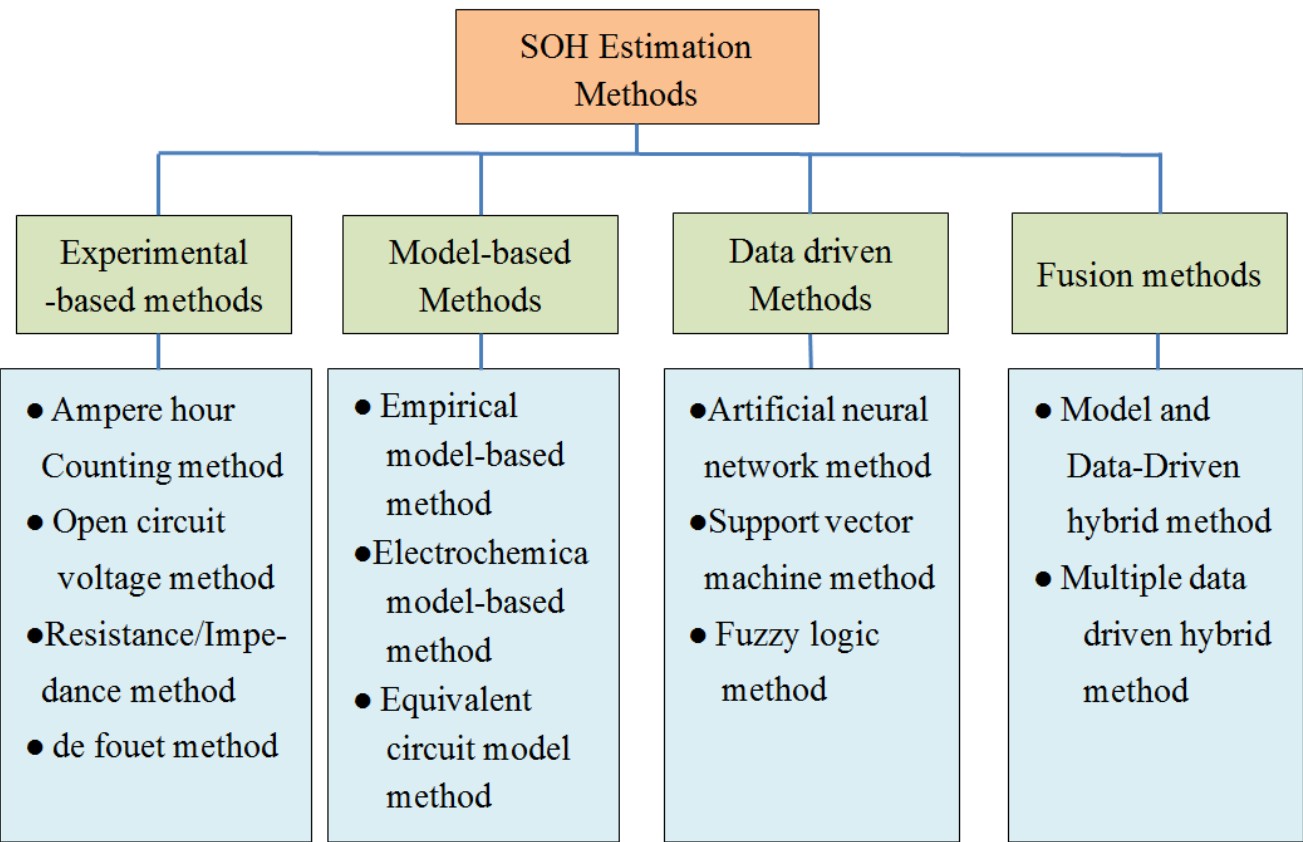

**Figure 2.** State of health (SOH) estimation methods.

### 4.1. Experimentally Based Methods

Experimental methods are usually carried out in the laboratory because they require specific equipment and are time-consuming. Experimental methods estimate the SOH by collecting data and measurements that can be used to understand and evaluate the battery aging behavior. The experimental methods usually require less computation and are easy to implement. Therefore, these methods are among the earliest methods used to estimate the SOH of VRLA batteries [23].

### 4.1.1. Ampere-Hour Counting Method

The ampere-hour counting method is one of the classical methods to estimate the battery SOH [32]. The common procedure of this method is to measure the present maximum capacity of the battery. In order to measure the current maximum capacity of the battery, the battery is first fully charged and then fully discharged; the current of the battery is then

recorded. Then, the maximum capacity of the battery can be calculated by integrating the discharge current.

$$C_{curr} = \int_{t_1}^{t_2} I dt \tag{5}$$

where I is the discharging current, and $t_1$ and $t_2$ are the starting and ending times of the discharge process, respectively.

The initial maximum capacity of a battery ($C_{initial}$) is usually provided by the manufacturer (referred to as nominal capacity); then, the SOH is determined using Equation (5).

The ampere-hour counting method is easy to implement under experimental conditions, and its estimation result is usually regarded as the true value of SOH, which can be used to verify the accuracy of other SOH estimation methods. At present, in power plants and substations, the ampere-hour discharge method is used to check the SOH of the battery pack every 1–2 years. When maintenance staff find a failed battery, they should replace it immediately to maintain the battery in good working condition. The ampere-hour counting method has some disadvantages. For example, it takes too long to test for the battery to be fully charged and discharged, so it is not suitable for online SOH estimation. The full discharge test is also harmful to the battery because deep discharge shortens the service life of the battery [33].

### 4.1.2. Open-Circuit Voltage Method

The open-circuit voltage (OCV) of the battery has long been known to have a functional relationship with the battery SOH. If the open-circuit voltage of the battery is measured, the battery SOH can be estimated [34].

James H. Aylor et al. [35] proposed a new technology for estimating battery SOH. The technique employs coulometric measurement under loading conditions and open-circuit voltage under no-load conditions in order to predict the change of the battery SOH. This technique was developed to enhance the accuracy and to reduce the required rest period of open-circuit voltage measurement.

Mchrnoosh Shahriari [36] presented an online method for the estimation of the state of health (SOH) of VRLA batteries based on the state of charge (SOC) of the battery. The SOC is estimated using an extended Kalman filter and a neural network model of the battery. Then, the SOH is estimated online based on the relationship between the SOC and the battery open-circuit voltage using fuzzy logic and the recursive least squares method. Experimental results show good estimation of the SOH of VRLA batteries.

The open-circuit voltage of the battery cannot be directly detected in the floating charge mode. In order to accurately measure the open-circuit voltage of the battery, it is necessary to keep the battery offline for a long time to reach a stable state. In addition, in order to improve the estimation accuracy of the open-circuit voltage method, it needs to be used in combination with other methods.

### 4.1.3. Resistance/Impedance Method

The internal resistance of the battery is considered an important index of SOH because it is seriously affected by the degradation of battery performance. When the SOH of the battery decreases, the internal resistance increases. With the increase in internal resistance, the SOH of the battery decreases. Considering the strong correlation between internal resistance and SOH, internal resistance is regarded as a good tool to estimate SOH [37]. The two main methods used to evaluate battery SOH are the internal resistance method and the electrochemical impedance method [38].

The internal resistance method usually establishes the corresponding relationship between the internal resistance and SOH and then evaluates the battery SOH according to internal resistance. To measure the internal resistance, a sudden current change ($\Delta I$) is exerted on the battery, and the consequent voltage change ($\Delta U$) is measured. The internal resistance can be calculated as $R = \Delta U / \Delta I$. The next step is to perform a regression analysis

of the resistance/impedance and SOH. Finally, using the regression function, The SOH of the batteries is estimated [23].

The internal resistance method only needs to obtain the voltage and current, making it suitable application in online estimation of battery SOH. Generally, the internal resistance of the battery has a certain relationship with SOC and SOH, and maintenance personnel can use these relationships to monitor the battery status in real time [39]. However, due to the uncertainty of the relationship between internal resistance and SOC, the error of SOH estimation is slightly larger [40]. In addition, when the capacity of a lead-acid battery is greater than 60%, the internal resistance changes slightly. Therefore, the internal resistance method is only used to roughly judge the battery SOH.

EIS is a kind of electrochemical measurement method whereby a low-amplitude sine wave voltage (or current) disturbance signal is imposed on the battery. EIS has no effect on the internal state of the battery and provides more rich information on electrode process dynamics and electrode interface structure details than other conventional electrochemical methods. Based on the circuit model, the relationship between the EIS curve and SOH can be established to accurately analyze the SOH of the battery. However, EIS measurement requires sophisticated and professional test equipment, which has high requirements for the test environment. As the circuit model itself is a technical difficulty, the process of EIS measurement and SOH calculation is relatively complex, which leads to the time-consuming and high cost of the impact method to estimate SOH. Therefore, a simpler and more general method for obtaining EIS parameters online requires further research.

### 4.1.4. Coup de Fouet Method

After being fully charged, the battery is discharged with a constant current. In the first few minutes, the discharge voltage reaches the peak voltage and then rises to the discharge platform voltage [41–43]. This phenomenon is referred to as coup de fouet (Figure 3).

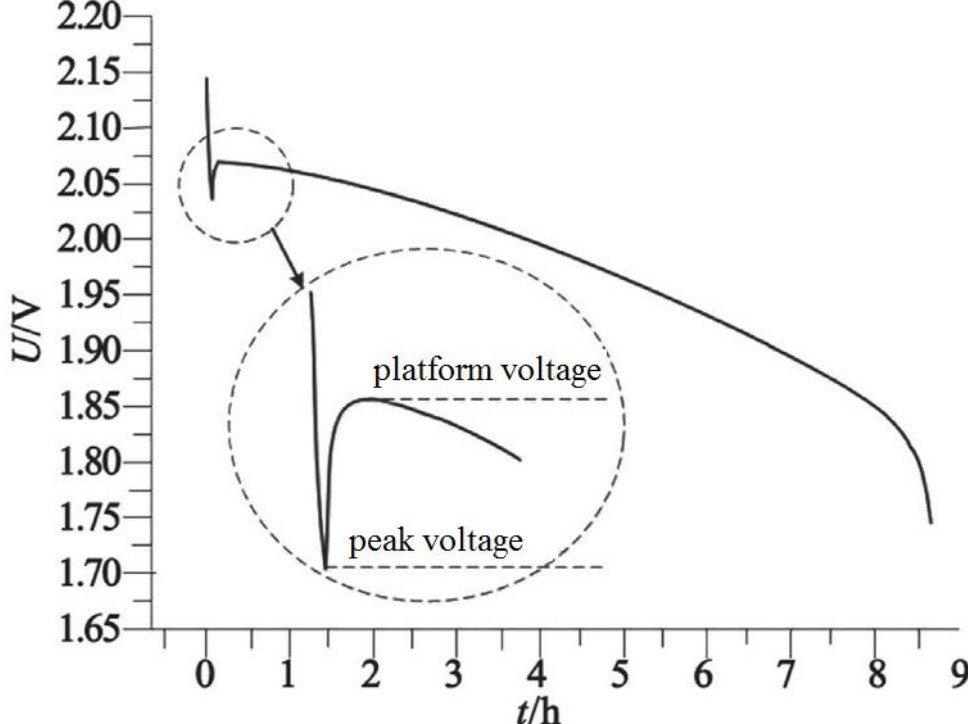

**Figure 3.** Coup de fouet of a lead-acid battery.

Several studies have applied the "coup de fouet" phenomenon to estimation of battery SOH. Phillip E. Pascoe et al. [44] found that the valley voltage and peak voltage in the coup de fouet phenomenon are linearly related to the actual available capacity of the battery; therefore, the SOH can be estimated according to the peak voltage and the platform voltage. A series of experimental studies revealed that the discharge rate and temperature have effects on the peak voltage and the platform voltage. Yuan et al. [45] assumed that the peak voltage and plateau voltage would be impacted under different discharge conditions (temperature and discharge rate). According to the coup de fouet phenomenon of the battery, the SOH is taken as the output variable, with the peak voltage, plateau voltage, discharge rate and temperature as input variables; accordingly, a battery SOH estimation model based on a BP neural network was built. The results show that the model based on a BP (backpropagation) neural network can effectively predict battery SOH. Due to the short discharge time during the test, the current working state of the battery is not be affected, and SOH can be estimated online. Compared with the traditional discharge test method, the coup de fouet method is more convenient and efficient and is very suitable for online detection of battery SOH as a backup power supply.

### 4.2. Model-Based Methods

Model-based methods use indirect measurement methods to predict the SOH of the battery. Empirical models, electrochemical models and equivalent circuit models can be applied.

### 4.2.1. Empirical Model-Based Method

The experience model-based method is used to simulate the aging process of the battery and test the effects of temperature, discharge depth, charge and discharge current on the battery life. The equivalent circuit model or a mathematical model are established with temperature, charging/discharging current and voltage as independent variables and SOH as a dependent variable. Then, the SOH and other battery parameters are calculated based on the model parameters.

Empirical models include the impedance empirical model and the capacity empirical model. The empirical model is used to first test the impedance (or capacity) of batteries in different life stages; then, the change trend of battery impedance (or capacity) over the whole life is obtained. Finally, the SOH of the battery is estimated according to the relationship between impedance and capacity [46].

John Wang et al. comprehensively studied the influence of battery temperature, discharge rate and SOC on battery capacity decay and established an Arrhenius model of capacity decay under the combined influence of temperature and discharge rate.

Compared with the equivalent circuit model, the empirical model is much more complicated, and it can explain many battery phenomena that the equivalent circuit model cannot. However, the empirical model lacks an explanation of the corresponding physical meaning, and its reliability and accuracy of estimating battery SOH often depend on the authenticity of the obtained experimental data. As a result, this method is not very common for SOH estimation.

### 4.2.2. Electrochemical Model-Based Method

The electrochemical model-based method is also called the battery mechanism model. In the electrochemical model-based method, a series of partial differential equations and algebraic equations is used to describe the physical and chemical processes inside the battery. The parameters in the model can represent the physical and chemical characteristics of the battery and can accurately reflect the changes thereof. Therefore, the electrochemical model-based method can be applied to the analysis of the VRLA performance decay mechanism and SOH estimation and provide theoretical support for prediction of the remaining life of batteries [47–49].

Chao Lyu et al. [50] presented a new method for battery SOH prediction by incorporating an electrochemical model into the particle filtering framework. A simplified electrochemical model of a lead-acid battery was introduced based on the theory of porous electrodes and the theory of diluted solution, which involve the charge conservation, electrode dynamics, liquid phase diffusion, liquid phase equilibrium and potential equilibrium of the solid phase. Figure 4 shows the schematic diagram of lead acid battery. The experimental results show that the model has the advantages of fast calculation speed, minimal damage to the battery and short detection time, making it suitable for SOH estimation and residual life prediction of backup VRLA battery packs in power systems.

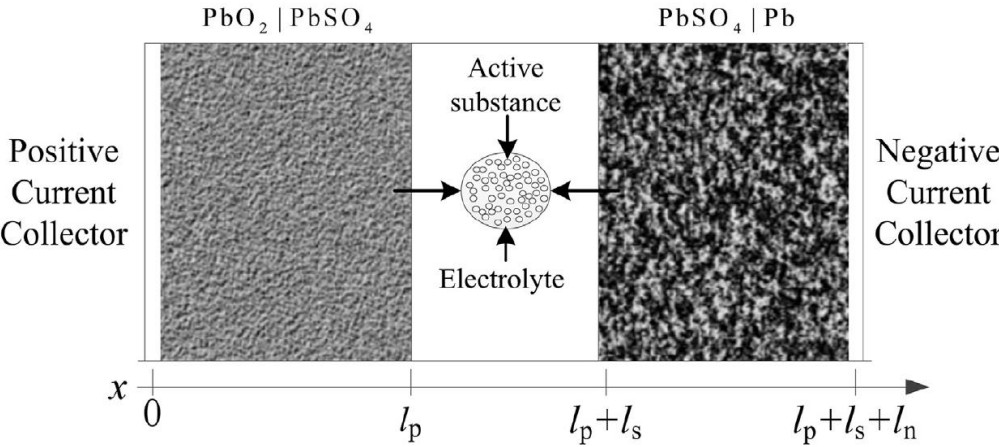

**Figure 4.** Schematic diagram of a lead-acid battery [50]. Reprinted with permission from Ref. [50]. 2016, Elsevier.

Evaluation of battery SOH using an electrochemical model has the advantages of clear physical meaning, high accuracy, universality, etc.; however, an electrochemical model includes many internal parameters of the battery, the control equation is complex, and many calculations are required, so its practical application is difficult.

### 4.2.3. Equivalent Circuit Model Method

The equivalent circuit model (ECM) is widely used in battery management systems because it comprises few parameters and is a simple mathematical model [51]. The ECM is composed of a voltage source, inductance, resistance, capacitor and other circuit components, which describe the charging and discharging characteristics of the battery through different combinations and simulate the dynamic characteristics of the battery. The general equivalent circuit model is shown in Figure 5. Generally, the basic equivalent circuit models of lead-acid batteries include the Rint model, the Thevenin model, the second-order RC (resistor–capacitor) model, etc. [52–55]. In the equivalent circuit model, a battery equivalent circuit is first established, and the model parameters are identified to estimate the SOH using algorithms, such as the least square method, Kalman filtering and artificial neural network.

According to the chemical principle of VRLA, Zhang et al. [56] established an equivalent model of a second-order reactive RC circuit to simulate the charging and discharging process. Based on the model, the open-circuit voltage and internal resistance of VRLA could be derived. The parameters of the model are estimated by the least recursive double algorithm with forgetting factor through system identification. Analysis and comparison show that the error of battery parameters identified by this algorithm is very small, and the second-order RC model simulation of the VRLA battery charging and discharging process is very accurate and efficient. Zhang et al. [57] built a second-order RC circuit model and used the terminal voltage comparison method to verify the rationality of the circuit model and the accuracy of identification parameters. In the model, the relationship between the ohmic internal resistance ($R_0$) and SOH is established, and the SOH of the

battery is estimated. Experimental verification shows that the relative error between the SOH of a VRLA battery estimated by the model and the SOH determined by the definition method is about 3%.

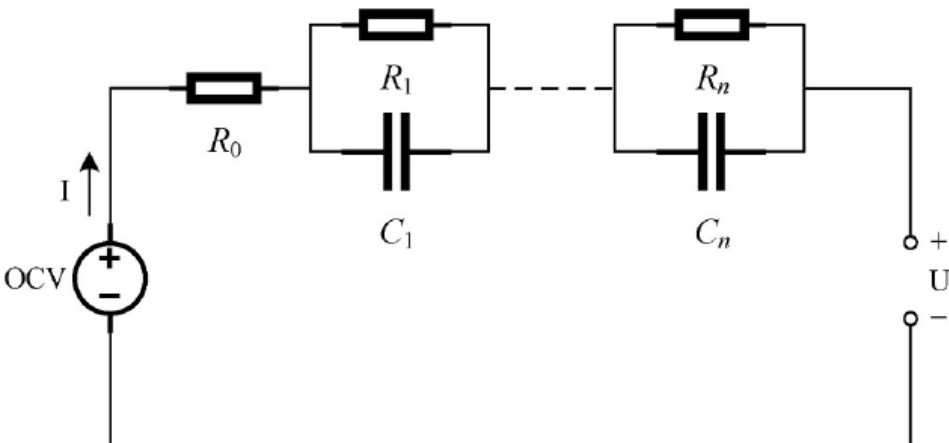

**Figure 5.** General equivalent circuit model [23]. Reprinted with permission from Ref. [23]. 2021, Elsevier.

The ECM is a model for the external characterization of VRLA batteries. The ECM has advantages such as concise structure, easy parameter identification, simple calculation processes and clear physical meanings and is therefore widely used in SOC and SOH estimation [58]. Because the equivalent circuit model often needs approximate equivalent treatment, when the key parameters of the battery cannot be obtained, some model prediction errors are expanded.

### 4.3. Data-Driven Methods

With the rapid development of big data and machine learning technology, data-driven technology has broken through the shackles of complex nonlinear systems that are difficult to model and has become the main research direction of battery health. Data-driven methods include artificial neural networks, support vector machines and Gaussian process regression. The general flow of the data-driven prediction method is shown in Figure 6 [59]. First, a large amount of battery information (such as voltage, current, temperature and impedance) is collected, which may come from past historical data or real-time data. Secondly, the battery degradation characteristics are extracted. The third step is to train a machine learning model to showing the relationship between the extracted characteristics and the SOH of the battery. Finally, once the machine learning model is determined, it is applied to evaluate the battery SOH.

4.3.1. Artificial Neural Network Method

An artificial neural network (ANN) is a network formed by the interconnection of neurons in a certain way, and a prediction model is obtained by training the thresholds and ratios of neurons through a large amount of data. Its typical structure consists of an input layer, a hidden layer and an output layer, as shown in Figure 7. Common neural networks for predicting battery SOH include BP neural networks, Elman neural networks and RBF (radical basis function) neural networks [60,61].

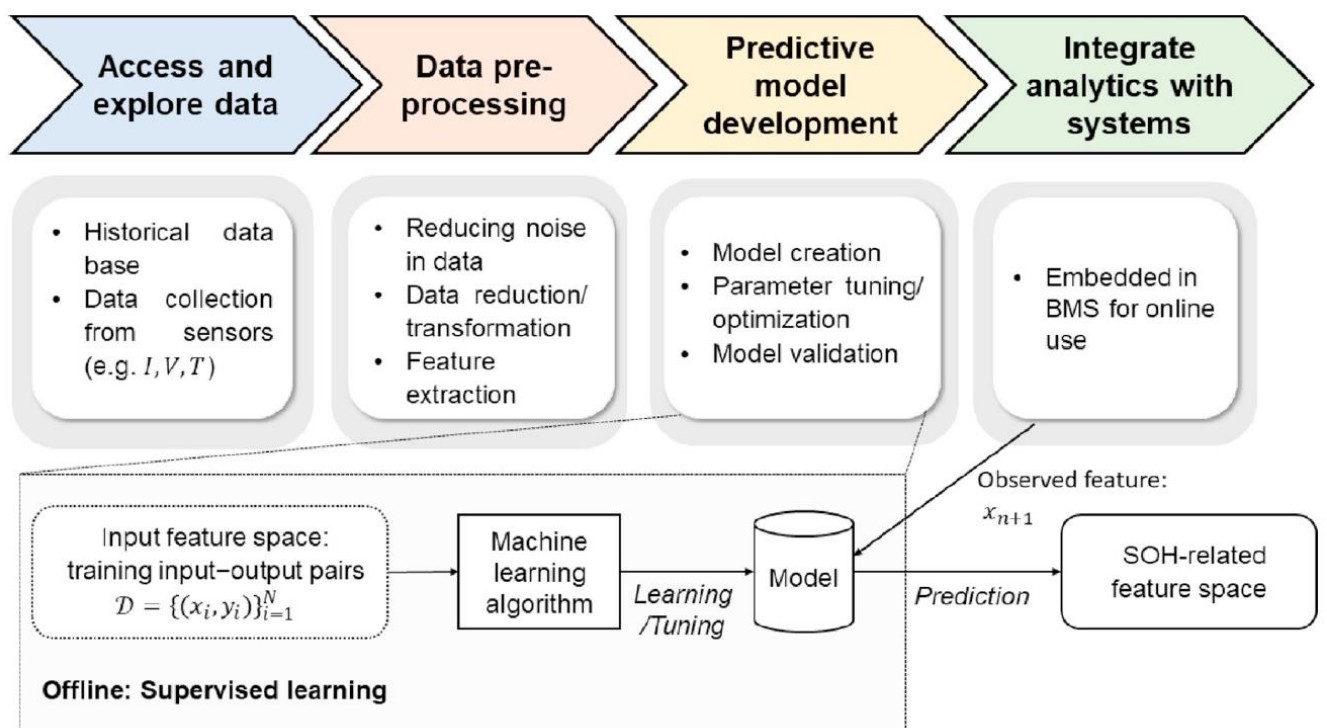

**Figure 6.** Generic workflow for data-driven SOH estimation methods [59]. Reprinted with permission from Ref. [59]. 2019, Elsevier.

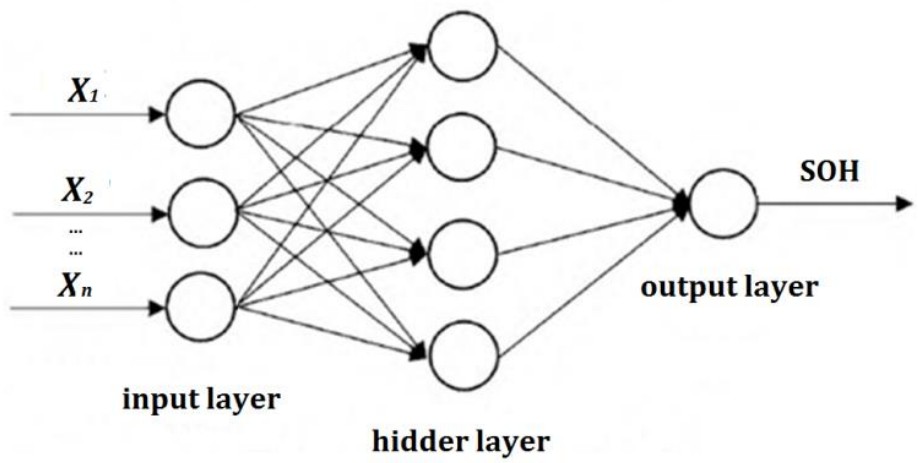

**Figure 7.** Structure of the BP neural network for model.

Talha et al. [62] proposed a simplified method to estimate the SOC and SOH of VRLA battery online using NN. Firstly, Terminal voltage ($V_t$) and open-circuit voltage (OCV) were measured for different charging currents ($I_{ch}$) and discharging currents ($I_{dch}$). $I_{ch}/I_{dch}$ and $V_t$ were used as NN input variables and OCV was output. Then, the SOC was calculated using an empirical function that described the relationship between SOC and OCV. Finally, the slope of SOC and current were used as training inputs, and the SOH was estimated by the NN. Mei et al. [63] built a wavelet neutral network (WNN) model for prediction of the operating life of a substation VRLA battery. First, a WNN model of battery operating life was established. Then, the experimental data were trained to obtain a WNN model for battery operating life prediction. The predicted results of the WNN model and actual data were compared. The experimental results show that the average relative error of prediction

is only 1.49%. The WNN model can quickly and accurately predict the working life of batteries in substations.

An ANN has the advantages of self-organization, self-adaptation, fault tolerance, self-learning evolution, high prediction accuracy and the ability to be applied to a variety of nonlinear prediction fields. However, the prediction ability of an ANN algorithm for a small dataset is poor, and the estimation error is affected by the training data.

### 4.3.2. Support Vector Machine Model

As a supervised machine learning algorithm, support vector machine (SVM) is used to solve classification and regression problems. Support vector machine constructs a hyperplane or a group of hyperplanes in high-dimensional or infinite dimensional space, which maximizes the isolation edge between positive and negative examples. Its principle diagram is shown in Figure 8. Similar to ANN, SVM is usually used to determine the relationship between input characteristics and SOH [23].

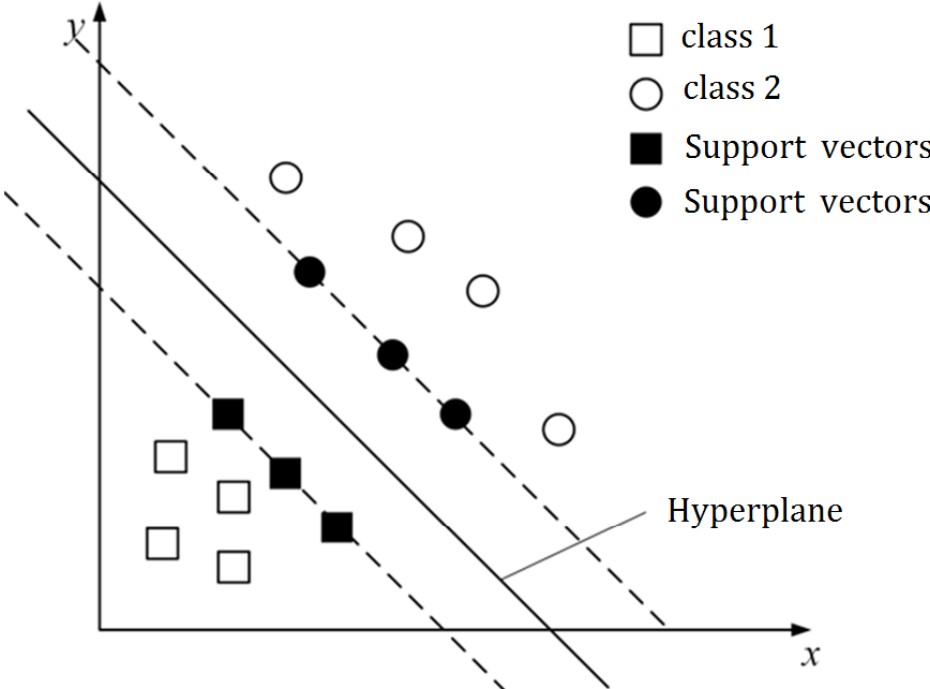

**Figure 8.** Principle diagram of SVM.

In view of the battery damage caused by checking discharge in substations, Cao [64] proposed a support vector machine algorithm to assess the health of substation batteries. In the study, the voltage, internal resistance, cycle times and activation time of a substation battery pack were taken as the feature vectors of the evaluation model. The experimental results show that the classification accuracy of the model is as high as 97.45%. Chang et al. [65] presented an approach using a Kalman filter (KF) based on SVM to estimate the battery SOH. The basic idea of a Kalman filter based on SVM is that the SVM time series model is first formed based on the measured data, and Kalman filter detection is combined with SVM prediction to estimate the SOH. Experimental results show that the estimation error of SOH was below 3%.

### 4.3.3. Fuzzy Logic Method

A complete fuzzy logic system (FLS) includes four main components, namely fuzzification, a knowledge database, a rules processor and defuzzification. The structure of a FLS is illustrated in Figure 9. The FIS involves four steps. First, fuzzification: the input values are fuzzified into the fuzzy language variables through the definition of a membership function. Second, knowledge database: a knowledge database is established to describe the membership functions for the input and output variables. Third, rules processor: according to the constraint conditions formulated by the rule base, a reasoning mechanism executes the inference procedure. Finally, defuzzification: the fuzzy output sets are converted to real output values [66].

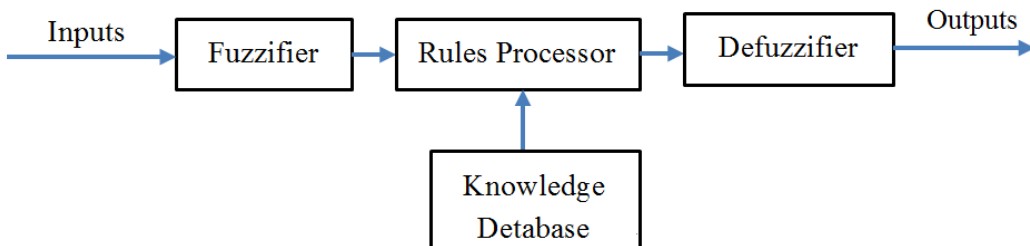

**Figure 9.** The structure of a fuzzy logic system.

When the fuzzy logic method is implemented in SOH estimation, the output of the fuzzy logic model is the battery SOH, and the inputs are extracted features that are related to the battery SOH. To implement this method, a rule base that describes how each extracted feature contributes to the SOH should first be built based on the training dataset. The rule base may be described by an expert or generated using neutral network algorithms [67]. Each set of data in the training dataset is a fuzzy set, and all the input values are then fuzzified into fuzzy membership functions. Next, the fuzzy output is calculated based on the rule base. Finally, according to the SOH of each fuzzy set, the estimated value of the SOH can be calculated using the weighted average of the SOH based on the fuzzy output [68].

A VRLA battery is a complex electrochemical system. With the charging and discharging process, there are many uncontrollable factors, such as the structural change of lead ions in electrolytes, the evolution of oxygen and hydrogen and the variation of ambient temperature, which affect the internal resistance, life and residual capacity of the battery. These uncontrollable factors bring many uncertainties and difficulties to VRLA battery SOH prediction, and the fuzzy logic system is just the means to solve the SOH estimation of this uncertain and complex electrochemical system.

Some researchers have applied the fuzzy logic method to estimate the SOH of VRLA batteries. FAN et al. [69] developed a fuzzy logic method to estimate the SOH of a VRLA battery in a substation online. According to the linear relationship between the amount of charge (Q) and the open-circuit voltage (OCV) of the battery, a fuzzy logic system based on Q-VOC slope and an SOH rule base was established. The input variable of the fuzzy system is the slope of the Q-VOC diagram, and the output is the SOH. The SOH of the battery was estimated in both online and offline states. The experimental results showed that the SOH error of online method was only 2% compared with the traditional check discharge method, demonstrating that this online SOH estimation method has the advantages of simplicity and short test time. Pritpal [70] described how impedance measurements, combined with fuzzy logic data analysis, have been used to estimate the SOH of lead-acid batteries. In the present method, the combination of fuzzy logic and EIS provided a powerful estimation method for SOH prediction of VRLA batteries. The fuzzy logic method is highly precise, offers good reliability and strong adaptability and can be used in a cheap microcontroller to provide low-cost cell surveillance systems.

*4.4. Fusion Methods*

In recent years, model fusion technology has received extensive attention from many researchers. The idea of the fusion method is to integrate multiple models, including experimental methods, model-based methods and data-driven methods, to give full play to their respective advantages and achieve accurate, reliable and robust battery health state estimation. Fusion-based methods usually include different model-based mutual integration, the merging of model methods and data-driven methods and the convergence of different data-driven methods.

Zhong et al. [71] proposed SOH estimation based on a fusion model for lead-acid batteries used in substations. Two models were established to estimate the SOH of a VRLA battery. The first model evaluates the relationship between the average resistance and SOH. The other model assesses the decline rate of battery voltage and SOH. According to the proportion of the influence of the nuclear discharge and floating charge state on SOH, a fusion model was established to estimate the SOH of a lead-acid battery in a substation. An accelerated life test was used to verify the proposed arithmetic, and the experimental results showed that the arithmetic was accurate and reliable and can realize real-time estimation the SOH of lead-acid batteries used in substations. Therefore, timely detection of poor SOH can greatly improve the safety and reliability of the battery pack.

The fusion estimation method overcomes the shortcomings of the single-model method or data-driven method, such as low prediction accuracy, poor reliability or misjudgment. It is an important method for battery SOH estimation in the future and has good application prospects.

## 5. Conclusions

Due to the complexity of electrochemical systems, the accurate estimation of SOH for VRLA batteries is still a challenge. Scholars at home and abroad have carried out a lot of research on the estimation of battery SOH and made many research advancements. Some research methods have been preliminarily applied. However, there is still a lack of a more complete theoretical system for battery SOH prediction, especially under actual operating conditions of backup power supply.

In this paper, the latest developments of SOH estimation methods for VRLA batteries in power system were reviewed. The basic principles, advantages and disadvantages of various methods were introduced. The main SOH estimation methods include experimental methods, model-based methods, data-driven methods and fusion methods. The traditional single model has poor accuracy in estimating battery SOH, and some SOH estimation methods can only be obtained offline and therefore cannot meet the future demand for high-precision and rapid battery SOH estimation in smart power plants and substations. For instance, although the ampere-hour counting method is simple, deep discharge damages the battery to some extent. The open-circuit voltage method requires a long standing, making it difficult to apply in the online estimation environment. The data-driven method can avoid the establishment of complex battery models. However, the establishment of databases, the selection of characteristic factors and the updating of estimation models are also considerable challenges. At present, the combination of model-based methods and data-driven methods has been widely used and achieved remarkable results. In particular, fusion methods are accurate and reliable for practical applications for electric power backup lead-acid batteries, effectively avoiding the adverse effects of systematic error and accidental error on SOH estimation.

The future development of VRLA battery SOH estimation may focus on the following aspects. First, in order to improve the accuracy of the model, efficient and accurate parameter identification methods should be further studied and developed. Second, various factors affecting battery SOH should be considered. By comparing the size of various factors, a more accurate battery model should be established to achieve accurate SOH estimation. Finally, with the development of artificial intelligence and big data technology, the integration of data and models, the complementary coordination of offline and

online methods and the realization of online application requirements will also be also an important research direction to improve battery SOH estimation.

**Author Contributions:** Conceptualization, R.Y.; methodology, C.H.; validation, H.W.; investigation, Y.M. and L.X.; resources, G.L.; writing—original draft preparation, H.W.; writing—review and editing, C.H.; visualization, Y.M.; supervision, R.Y.; project administration, G.L.; funding acquisition, L.X. All authors have read and agreed to the published version of the manuscript.

**Funding:** This research was funded by the science and technology project of Zhejiang Zheneng Jiahua Electric Power Generation Co., Ltd. (ZNKJ-2021-041).

**Institutional Review Board Statement:** Not applicable.

**Informed Consent Statement:** Not applicable.

**Data Availability Statement:** Not applicable.

**Conflicts of Interest:** The authors declare no conflict of interest.

## Abbreviations

| | |
|---|---|
| AC | alternating current |
| ANN | artificial neural network |
| BP | backpropagation |
| DC | direct current |
| DVA | differential voltage analysis |
| ECM | equivalent circuit model |
| EIS | electrochemical impedance spectroscopy |
| FIS | fuzzy logic system |
| ICA | incremental capacity analysis |
| ICP | Inductively coupled plasma technology |
| KF | Kalman filter |
| OCV | open-circuit voltage |
| PAM | positive active material |
| RC | resistor–capacitor |
| SEM | scanning electron microscope |
| SOC | state of charge |
| SOH | status of health |
| SVM | support vector machine |
| VRLA | valve-regulated lead acid |
| XRD | X-ray diffraction technology |
| WNN | wavelet neutral network |

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
