# Peer review of "Review of Degradation Mechanism and Health Estimation Method of VRLA Battery Used for Standby Power Supply in Power System"

_coatings, doi:10.3390/coatings13030485_

Round 1

Reviewer 1 Report

Dear authors, 

The review article "Review of Degradation Mechanism and Health Estimation  Method of VRLA Battery Used for Standby Power Supply in Power System" is not acceptable for publication. The reasons are 

1 This review is not suitable for coating journal as mentioned in the aim and scope "science and engineering of coatings, thin and thick films, surfaces and interfaces"

2. In depth analysis missing for both Degradation Mechanism and Health Estimation of VRLA.

3. No detailed literature review is presented, and this can be evidenced by <50 cited references

4. Most figures are general in nature and not specific to the particular method reported in the literature. 

5. No copyright information in given for figures adopted  

6. Figure 2 some text is got cut (ex: method)

7. Challenges and perspectives should be discussed for a review article, which was missing in this article. 

Author Response

Dear reviewer, 

 Thank you for your painstaking and detailed analysis of our review paper. We are highly grateful for the dedication with which you have read our paper. We have tried our best to overcome the shortcomings which were highlighted by the reviewer and hope to satisfy the reviewer's concerns.

    Best Regards,

Hu Chen

-----------------------------------------------------

China Electric Power Research Institute

Address: No. 15 Xiaoying East Road, Qinghe, Beijing 100192, China

Reviewer 2 Report

Dear Authors,

Thank you for submitting your work to Coatings. Your summary of the standby operation and failure mechanisms of backup VRLA with the review of SOH estimation methods is a useful addition to the literature.  I recommend some editing to make the manuscript more useful.  Please state what is required for online monitoring and assessment explicitly. If nonintrusive monitoring is possible for any of the operating modes is possible, please identify them; if not, please define a “tolerable disturbance”, i.e., a short, periodic disruption of operations that can enable effective online monitoring.  

When describing the methods, please rank them relative to their applicability and consider moving “lab methods” to the appendix.  More details on required measurements for data-driven methods would be very helpful.

Additional comments are given as follows

o   Introduction

§  DC is obvious, but the acronym was not spelled out

§  References in the bibliography do not include the names of all authors

§  Past tense should be used when describing examples of grid downtime caused by lead-acid storage failures

§  Please identify and cite any similar review articles and then state the differentiating aspect of your paper explicitly.

o   Operating characteristics of standby VRLA

§  Figure 1 is constructive, but please describe floating charging, equalized charging, and CV/CC charging. Note any opportunities (and limitations!) related to online estimations of SOH. Can any of the SOH methods be applied to any of these regimes?

o   Failure mechanism in VRLA

§  PAM (Positive active material) acronym not defined

§  Please consider adding a comment on how failure modes can be detected using electrical testing

o   Definition of SOH

§  Continued sentences (paragraphs starting with “where”) after equations should not be indented.

§  The text following Eq. (2) is the same as that following Eq. (1). It should identify resistances R_new, etc.   In addition, please define SOC and specify the SOC conditions used for evaluating SOH using resistive measurements (the internal resistance is a function of SOC).

o   SOH estimation methods

§  In Figure 2, the word “method” of “equivalent circuit method” was cut off

§  Given the objective of monitoring VRLAs in operation, it is essential to discuss the usefulness of different methods. For example, how useful are laboratory methods? Are there any limitations? 

§   Consider reorganizing the text so that the most useful methods (relative to online monitoring) are discussed in detail and presented in descending order of their applicability for online estimation. The methods not directly applicable can be relegated to an appendix

§  ECM (equivalent circuit model) -- acronym not defined.  The final sentence of ECM contradicts prior statements. More details of requisite measurements needed to extract the ECM model would be helpful.

o   Data-driven methods

§  Please provide what type of data needs to be collected.  Do any of the charging states naturally provide data for this type of modeling? If not, please state explicitly what specific data was used to build models. How was impedance measured?  Are these inputs readily measured or required for interrupted operations?

§  What is meant by “self-learning evolution”?

Consider adding a list of abbreviations

Author Response

(The authors gave the same response as above.)

Reviewer 3 Report

Dear,

This is a current/modern research that involves serious concerns of the electrical sector for accurate estimation of SOH for VRLA batteries, especially in power plants and substations, which as a result, some switches cannot function normally in time and the failure cannot be isolated. , which eventually leads to the accident of total loss of voltage in the substation causing inconvenience. The search for increasingly accurate estimation methods is still a challenge for the area. The authors were happy for the search for the best methods. As the authors themselves reported, especially the fusion methods are accurate and reliable in the practical application of electrical power backup lead-acid battery, which can effectively avoid the adverse effects of systematic errors and accidental errors in estimating SOH. I am satisfied with the development of the research, the methodology used, as well as the results and conclusions. In Brazil, some battery manufacturing companies actually use online techniques to measure the health of batteries for use in substations, and thereby reduce measurement errors and negative impacts. Some very current works that can contribute to the bibliographical review of the research, and that were works from battery manufacturers are in these references and I believe that they will give greater prominence to the originality and relevance of the studies developed, are these:

1) Nascimento, R.; Ramos, F.; Pinheiro, A.; Junior, W.d.A.S.; Archangel, A.M.C.; Filho, R.F.D.; Mohamed, M.A.; Marinho, M.H.N. Case Study of Backup Application with Energy Storage in Microgrids. Energies 2022, 15, 9514. https://doi.org/10.3390/en15249514

2) by Araujo Silva Júnior, W.; Vasconcelos, A.; Archangel, A.C.; Costa, T.; Birth, R.; Pereira, A.; Jatoba, E.; Filho, J.B.; Barrett, E.; Dias, R.; Marinho, M. Characterization of the Operation of a BESS with a Photovoltaic System as a Regular Source for the Auxiliary Systems of a High-Voltage Substation in Brazil. Energies 2023, 16, 1012. https://doi.org/10.3390/en16021012

3) Costa, T.; Archangel, A.; Vasconcelos, A.; Silva, W.; Azevedo, C.; Pereira, A.; Jatoba, E.; Filho, J.B.; Barrett, E.; Villalva, M.G.; Marinho, M. Development of a Method for Sizing a Hybrid Battery Energy Storage System for Application in AC Microgrid. Energies 2023, 16, 1175. https://doi.org/10.3390/en16031175

Furthermore, congratulations to the authors for the quality of the research.

Author Response

(The authors gave the same response as above.)

Round 2

Reviewer 1 Report

Dear authors,

The revised version can be accepted.